# Bacterial Cellulose—Carboxymethylcellulose Composite Loaded with Turmeric Extract for Antimicrobial Wound Dressing Applications

**DOI:** 10.3390/ijms24021719

**Published:** 2023-01-15

**Authors:** Gabriela Isopencu, Iuliana Deleanu, Cristina Busuioc, Ovidiu Oprea, Vasile-Adrian Surdu, Mihaela Bacalum, Roberta Stoica, Anicuţa Stoica-Guzun

**Affiliations:** 1Department of Chemical and Biochemical Engineering, Faculty of Chemical Engineering and Biotechnologies, University “Politehnica” of Bucharest, 1-7 Polizu Street, 011061 Bucharest, Romania; 2Department of Science and Engineering of Oxide Materials and Nanomaterials, Faculty of Chemical Engineering and Biotechnologies, University “Politehnica” of Bucharest, 1-7 Polizu Street, 011061 Bucharest, Romania; 3Department of Inorganic Chemistry, Physical Chemistry and Electrochemistry, Faculty of Chemical Engineering and Biotechnologies, University “Politehnica” of Bucharest, 1-7 Polizu Street, 011061 Bucharest, Romania; 4Academy of Romanian Scientists, 3 Ilfov Street, 050045 Bucharest, Romania; 5Department of Life and Environmental Physics, Horia Hulubei National Institute of Physics and Nuclear Engineering, 30 Reactorului Street, 077125 Măgurele, Romania; 6Department of Anatomy, Animal Physiology and Biophysics, Faculty of Biology, University of Bucharest, 91-95 Splaiul Independentei Street, 050095 Bucharest, Romania

**Keywords:** bacterial cellulose, carboxymethyl cellulose, turmeric extract, wound dressing

## Abstract

Bacterial cellulose (BC) is a biopolymer whose properties have been intensively studied, especially for biomedical applications. Since BC has no antimicrobial activity, it is necessary to use bioactive substances for developing wound healing applications. Another drawback of BC is the loss if its water retention capacity after dehydration. In order to overcome these problems, carboxymethyl cellulose (CMC) and turmeric extract (TE) were selected for the preparation of BC composites. Citric acid (CA) was used as the crosslinking agent. These composites were tested as potential antimicrobial wound dressing materials. TE-loaded BC–CMC composites were characterized in terms of their morphology, crystallinity, and thermal behavior. Swelling tests and curcumin-release kinetic analysis were also performed. All the composites tested had high swelling degrees, which is an advantage for the exudate adsorption from chronic wounds. The antibacterial potential of such composites was tested against *Escherichia coli* (*E. coli*), *Staphylococcus aureus* (*S. aureus*), and *Candida albicans* (*C. albicans*). The in vitro cytotoxicity toward L929 fibroblast cells was studied as well. The obtained results allow us to recommend these composites as good candidates for wound dressing applications.

## 1. Introduction

Skin is the largest organ of the human body, and it acts as a physical barrier against the external environment, protecting us from harmful factors. This is just one of the skin’s functions among others, like thermoregulation, endocrine and exocrine activity, immunological role, as well as the creation of heat, cold, touch, and pain sensations through specific receptors. The protective function of the skin could be perturbated by different injures caused by external or internal factors. Wounds can be classified using many criteria, the most important among them being the process and the duration of healing. In this sense, we can mention acute and chronic wounds [1]. Wound healing in the case of acute wounds is more rapid than for chronic ones, the latter being more difficult to heal. Wound healing is a complicated process whose main stages are inflammation, proliferation, and maturation, being largely described in literature [2,3]. The mechanism of the self-healing of a wound is also involved in this process, but it is slow and susceptible to external infections. For these reasons, wound dressings provide an invaluable help to quicken the healing process [2]. An optimal dressing must fulfil a great number of conditions, such as: maintaining humidity of the wound site, removing excess exudates, preventing microbial infections, and ensuring oxygen exchange. Being a material in contact with the skin, the dressing must be non-toxic, non-allergenic, and comfortable. From an economic and environmental point of view, dressings must be also cheap and biodegradable.

Two main categories of biomaterials have been intensively used as wound dressing: synthetic and natural ones, each category having certain advantages and disadvantages. The main advantages of synthetic materials are that they can be designated in a such way to meet the desired purposes (e.g., adhesion and biodegradability properties), they can be produced at an industrial scale, and their properties could be controlled by modifying the production parameters. Some examples of synthetic biodegradable polymers are: polycaprolactone (PCL), polylactic acid (PLA), polyglycolic acid (PGA), and related copolymers [4].

Natural biopolymers deserve special attention, being studied with the aim of overcoming the disadvantages of synthetic materials. The most well-known natural biopolymers used for wound dressing are collagen, gelatin, silk sericin, alginate, chitin, and chitosan, with collagen being by far the most studied [5,6].

Among the biopolymers that are gaining attention as possible candidates for wound dressing, we find bacterial cellulose (BC), a biopolymer obtained by microbial fermentation of non-pathogenic bacteria of the genus *Komagateibacter*, such as *K. xylinus* (synonyms: *Acetobacter xylinus*, *Gluconacetobacter xylinus*) [7,8]. Two arguments sustain the use of bacterial cellulose as wound dressing material: its intrinsic properties and its potential to be used in composite materials with or without chemical modification. Among the composite materials in which BC plays an essential role, the most important are hybrid materials with different nanoparticles and BC-based composites with different biopolymers (e.g., alginate, chitosan, collagen).

If one considers the intrinsic properties of BC, the following elements should be mentioned: an ultrafine 3D network structure and a high liquid-holding capacity (the water content of pristine BC could be up to 99%); thus, it could control wound exudate and provide enough moisture to help wound healing. It is biocompatible, and it has no cytotoxicity, but it presents a strong hydrophilicity, good mechanical properties, flexibility, and a satisfactory permeability for liquid and gases [9,10]. The disadvantages of BC are the lack of antimicrobial activity, a biocompatibility which is not at the highest level, and the loss of its rehydration capacity after drying. To overcome these problems, there are many possibilities which were already investigated through in situ and ex situ modifications [10,11]. The combination with organic antimicrobial substances, the impregnation of antibiotics on BC, or grafting of antibiotics onto modified BC are some examples. In addition, inorganic antimicrobial particles and carbon nanomaterials deposited on BC and BC-based composites could transform BC and BC-based composites into antimicrobial materials [2,12].

Many natural substances have been used to enhance the antimicrobial activity of BC including thymol, sesamum oil, papain solution, and curcumin [13,14,15,16]. Since there are also many plant extracts with antimicrobial effect, the research in this regard will be continued [17].

Among these, curcumin should be highlighted due to its remarkable properties, being an anticarcinogenic, anti-inflammatory, antioxidant, antimicrobial, anti-coagulant, antimutagenic, and anti-infective agent [18]. Curcumin is also used in traditional medicines for various skin wounds [19,20].

There are many papers which report the use of curcumin in dressing materials entrapped in composites with different biopolymers and other antimicrobial substances. Some recent studies used curcumin cross-linked with chitosan–PVA membranes, curcumin and borneol-loaded polycaprolactone–gelatin nanofiber membranes, and alginate with (2-Hydroxy isopropyl)-β-cyclodextrin [21,22,23].

Among cellulose derivatives, carboxymethyl cellulose (CMC) (an anionic polysaccharide) has several physicochemical properties like biocompatibility, biodegradability, low toxicity, and nonimmunogenicity which make it attractive for wound dressing applications [24]. CMC is also a low-cost biopolymer produced mainly from cellulose derived from wood or from cotton linters. It could be also synthesized using cheaper cellulose sources as alternatives to wood cellulose [25]. Bacterial cellulose could be also used as raw material for CMC synthesis and there are only few studies about BC carboxymethylation [25,26,27]. Despite the advantage of the high purity of bacterial cellulose, there are several disadvantages to using BC to produce CMC. First, BC has a lower reactivity compared to wood cellulose conversion under comparable conditions. In addition, BC degradation could be observed during carboxymethylation in alkaline conditions. Finally, the cost of obtaining BC is still high.

Its low mechanical strength and its lack of bioactivity are some drawbacks of using CMC for wound dressing applications. Different natural substances and antimicrobial nanoparticles (especially silver and ZnO) were incorporated in CMC hydrogels or films for developing antibacterial materials [28,29,30].

Due to its hydrophilic nature, CMC offers the possibility to be blended with other biomaterials (poly-vinyl alcohol, poly-ethylene glycol, alginate, gelatin, collagen) and to prepare new wound dressing biomaterials [31,32,33,34]. CMC could also be used for obtaining biodegradable superabsorbent hydrogels which can absorb large water quantities, and which could have biomedical applications. The following substances could be used as crosslinking agents: epichlorohydrin, glutaraldehyde, and citric acid, the latter being used more and more because the other listed substances are toxic [33,35]. Not only CMC, but also BC could be crosslinked using citric acid to obtain hydrogels with biomedical applications, including wound dressing [36,37]. The carboxylate moiety of citric acid could react with the hydroxyl moiety of both BC and CMC to form crosslinks between the two molecules. CMC could be also deposited into a BC matrix without crosslinking to obtain BC–CMC reswellable hydrogels [38]. In most studies, BC–CMC composites are obtained by CMC addition into culture media (in situ method). These composites had a good swelling degree, but their crystallinity and mechanical strength decreased [38]. In our previous research papers, we have demonstrated that BC could be used in drug delivery systems as native membranes or as fibrils in composite materials with carboxymethylcellulose or with chitosan and poly (vinyl alcohol). As model drugs, BC was tested with ibuprofen sodium salt and amoxicillin [39,40,41].

Starting from these finding, the aim of the current study was to fabricate new composite materials using BC and CMC to improve the properties of these biopolymers and to demonstrate their efficiency as practical wound-dressing products with a high swelling capacity. At the same time, the use of CMC, which is a cheap biopolymer, could reduce the production cost of the composite materials.

Since BC and CMC have no antimicrobial properties, turmeric extract has been used mainly for its curcumin content, as the antimicrobial and antioxidant effects of curcumin are well known. The developed materials were characterized for their structural and physicochemical properties. Their antimicrobial and antioxidant activity and in vitro cytotoxicity toward L929 fibroblasts were also tested.

## 2. Results and Discussion

### 2.1. Characterization of BC–CMC Composites

Four samples were used in this study and their synthesis conditions and symbolic notations are depicted in Table 1.

Turmeric extract was obtained from turmeric powder in ethanol, through Soxhlet extraction. The composition of the extract was determined through Ultra-High-Performance Liquid Chromatography (UHPLC). The main components of the turmeric extract were: bisdemethoxycurcumin: 16.37%, demethoxycurcumin: 25.38%, and curcumin: 58.25%. The TE chromatogram is presented in Appendix A.

#### 2.1.1. Scanning Electron Microscopy

SEM images of the surface morphology of the studied samples are presented in Figure 1. The images of the BC–CMC composite without TE (P1 and P3) present the three-dimensional network of BC, which was impregnated only with CMC aqueous solution that was crosslinked after drying with CA. In the case of the composites containing TE (P2 and P4), the presence of the turmeric extract components, especially curcumin, changing the aspect of the SEM images, as one can observe from Figure 1 (P2b and P4b). Some aggregates were formed on the BC–CMC surface, these being more pronounced for the composite with a higher crosslinked agent concentration (P4b). Our results are similar to those reported by Tangsatianpan et al. (2020), with the remark that the aforementioned authors have tested BC loaded only with curcumin, without CMC and without a crosslinker agent [42].

#### 2.1.2. FTIR Analysis

FTIR spectra of the studied samples and some reference samples are presented in Figure 2. Figure 2a presents the spectra of CMC, the BC–CMC composite, and the BC–CMC composite in which CA was used as the crosslinking agent for CMC as reference samples. In the spectrum for BC-CMC, one can distinguish peaks at 3343 cm^−^^1^, which correspond to O-H stretching, and at 2894 cm^−^^1^, which are attributed to C-H stretching of -CH_2_ and are both characteristic to BC and to CMC. At 1032 cm^−^^1^, we observed a peak that can be attributed to C-O-C stretching vibrations of pyranose ring structure, while at 897 cm^−^^1^_,_ there is a peak characteristic of β-1,4-glycocidic linkages of the cellulose polymer [13,15,43,44]. From the CMC spectrum, one can observe characteristic bands at 1585 cm^−^^1^ (the peak of nonhydrated C=O groups) which confirm the presence of COO¯ groups, at 1420 cm^−^^1^ (assigned to –CH_2_ scissoring), and at 1020 cm^−^^1^ which represents carboxymethyl ether group stretching [45]. In the BC-CMC spectrum, there is an overlapping of the peaks due to very similar structures of the two biopolymers. Only the peak at 1585 cm^−^^1^ of CMC was visible in the two reference spectra (BC-CMC and BC-CMC-CA). The peak at 1735 cm^−^^1^, which was expected to arise in the spectrum of the BC–CMC composite in which CMC was crosslinked, was not visible, possibly due to the small quantity of CA used as crosslinking agent. In the spectra of the P2 and P4 samples, two peaks could be assigned to curcumin, one at 1278 cm^−^^1^ and another at 1523 cm^−^^1^. With the help of FTIR microscopy, we looked at the spatial distribution of curcumin in the composites P1–P4. The maps corresponding to 3480 cm^−^^1^, 1523 cm^−^^1^, 1278 cm^−^^1^, and 1161 cm^−^^1^ are presented for the samples P1–P4 in Figure 3, together with an image from microscope of the analyzed region.

The peak at 3480 cm^−1^, which is characteristic for the stretching vibration of the –OH moiety, is common to both components. The band that appeared around the 1161 cm^−1^ range was assigned to asymmetric stretching vibrations of C–O–C in the 1,4-glycosidic linkages of cellulose. The peaks from wavenumbers 1523 and 1278 cm^−1^ are specific only for curcumin. The peak at 1523 cm^−1^ was assigned to the bending vibration of olefin C–H bound to the benzene ring, while the peak at 1278 cm^−1^ was assigned to the C=O stretching vibrations attached to the aromatic nucleus.

The samples P1 and P3 exhibit similar FTIR maps at all the above wavenumbers, as there was no distribution gradient of a second component (curcumin). The P2 and P4 samples indicated clear differences between the FTIR maps from curcumin wavenumbers (1523 and 1278 cm^−1^) vs. FTIR maps from cellulose wavenumber (1161 cm^−1^). The FTIR maps from the common wavenumber 3480 cm^−1^ represent features from both types of compounds.

#### 2.1.3. XRD

Figure 4 shows the results for the XRD analysis for the studied samples. In the spectra of all the samples, one can find three strong Bragg peaks for type I cellulose observed at 14.4, 16.9, and 22.7 degrees. A question arises regarding the absence of the curcumin characteristic peaks, because it is known that free curcumin exhibits peaks between 5–30° as a high crystalline substance. It is possible that curcumin may have been entrapped in the BC–CMC composites in an amorphous form; this assumption is in agreement with other literature reports [46,47,48]. X-ray diffraction patterns were used to determine the samples crystallinity. The crystallinity values for the studied samples are: P1—46.54%, P2—50.54%, P3—50.31%, and P4—50.8%. These data are in agreement with those reported by Ojagh et al. (2021) for BC–CMC composites obtained in situ during BC fermentation [49].

#### 2.1.4. Thermogravimetric Analysis

The thermogravimetric curves for the studied samples (P1–P4) are presented in Figure 5. From these, one could observe common features for all curves (TG and DSC). The sample P1 lost 6.70% up to 200 °C, most probably due to the elimination of some adsorbed residual water molecules. The DTG curve indicates that this process was attaining the maximum speed at 98.0 °C. The associated effect was endothermic with a minimum at 60.1 °C (an explanation is given in Appendix A. The principal mass loss of 52.83% occurred between 200–375 °C and was accompanied by an exothermic effect which reached its maximum at 340.1 °C, while the maximum mass loss speed was obtained at 328.9 °C, as seen on the DTG curve. This step was associated with an oxidative-degradative process of the cellulose backbone. The burning of the cellulose residuals occurred between 375–502 °C, with the maximum rate at 461.6 °C, when there was recorded a mass loss of 34.26%. This process was accompanied by a strong exothermic effect on the DSC curve, reaching its maximum at 469 °C, indicating the oxidative nature of the reactions. On the DSC curve, there was a small shoulder observed at 544.3 °C, which can be assigned to the burning of the residual carbonaceous mass. The P2 sample which contains TE had a similar behavior during thermal analysis, its second exothermic effect being placed at higher temperatures (+10 °C), due to the thermal stability of the turmeric components. A similar shift (+17 °C) can also be observed between the P3 and P4 samples for the exo II position. The centralized data from thermogravimetric analysis are presented in Table 2 for all the studied samples. More data for each mass loss and thermal analysis were obtained, together with the subsequent software processing, on each sample (P1–P4) and are presented in the Appendix A, in order to clarify all the data included in Table 2.

#### 2.1.5. Swelling Studies

Swelling studies are very important for a wound dressing that must absorb the exudate as a condition to sustain the wound healing process, especially for chronic wounds. For the present study, the leading idea was to preserve the high swelling properties of BC hydrogels. As already shown, the tridimensional network of BC was destroyed after drying and the swelling capacity greatly decreased [37]. In the composition of the studied samples, the presence of CMC and the crosslinked process with CA play an important role in maintaining a good hydration capacity of the new wound dressing composites. The results of the swelling ratio for the samples P1–P4 in PBS (phosphate buffered-saline; pH = 7.4) at room temperature are depicted in Figure 6. The samples P1 and P2 were observed to have similar swelling degrees. The same observation can be stated for the samples P3 and P4. There is, however, a notable difference, between the studied samples regarding the concentration of the crosslinking agent (the pair P1–P2 versus the pair P3–P4). In the case of the increased concentration of CA, the swelling degree decreased. This could be explained by the degree of crosslinking, but for an extended discussion more experiments are necessary in order to analyze a wider concentration range of the crosslinked agent.

#### 2.1.6. Curcumin Release Studies

The curcumin release curves for the P2 and P4 samples are presented in Figure 7. Due to the fact that the release medium was a mixture of ethanol and PBS (60:40, *v*/*v*), the release was faster than in PBS. Therefore, after 100 min, the results indicated that more than 85% ± 1.25% of the loaded curcumin was released from both samples. Figure 7 shows a faster release for the P4 sample. When comparing the two samples, only the crosslinking agent concentration is different. For the sample with the higher CA concentration (P4), the release rate was also higher. For the moment, an explanation could be provided by the SEM images. In Figure 1/P4b, one can observe an agglomeration of curcumin particles on the surface of the sample, which could be favorable to the faster dissolution of curcumin in the release medium. For both samples, we observed a burst effect which, generally, must be avoided, because it can reduce the drug effect [50]. In our case, this burst effect could be produced by the curcumin adsorption on the sample surfaces. The Korsmeyer–Peppas equation, presented below (Equation (1)), was used to describe the curcumin release kinetics from the studied samples.
(1)$$\frac{M_{t}}{M_{\infty }}=kt^{n}$$

In Equation (1) *M_t_* is the amount of drug released at time *t* and *M_∞_* is the amount of drug released at ∞, practically at equilibrium, and *k* is a kinetic constant (time^−^^1^). The exponent n indicates the drug release mechanism. From the released curves depicted in Figure 7, we obtained *n* = 0.249 for the P2 sample and *n* = 0.239 for the P4 sample, which indicates a quasi-Fickian diffusion (*n* < 0.5). Our results are in accordance with those obtained by Tummalapalli et al. (2016), who studied the curcumin release from pectin and gelatin composites with curcumin [51].

#### 2.1.7. Antioxidant Activity

Oxidative stress is known to inhibit the wound healing process. This negative phenomenon could be avoided by releasing antioxidant substances. In our study, we chose curcumin as the antioxidant substance, which is one of the most important components of TE, and whose antioxidant properties are well documented [52]. The results obtained for the antioxidant activity of the BC–CMC composites with TE are presented in Table 3. As control samples, we also tested pure curcumin and turmeric powder, the latter being used to obtain the initial extract.

In comparison with turmeric and curcumin, both samples presented a moderate antioxidant activity, which was expected since the turmeric extract was used in small amounts during the sample preparation. A slightly higher antioxidant activity was observed for the P2 sample, which was crosslinked with a small quantity of CA. It is possible to increase the antioxidant properties of the samples by increasing the initial amount of turmeric extract and those of curcumin, but this could reduce the cell viability [53,54]. In order to reach an optimal curcumin content that would ensure both a high antioxidant activity and high cell viability, more studies are necessary.

#### 2.1.8. Antimicrobial Activity

The effects of curcumin on bacterial strains revealed in the literature are the followings: cell membrane disruptions, reactive oxygen species (ROS) induction, or cell division interruption, while intracellular acidification via inhibition of H^+^ extrusion was identified as a possible mechanism for inducing cell death in Candida species [18]. In this paper, the disc diffusion assay was applied because it is cheaper than other methods and it is easy to use, even if it has the disadvantage of dependence on the diffusion of the antimicrobial agent from the wound dressing sample into the agar or other culture medium.

Other methods, which are independent from the diffusion properties of the antimicrobial agent, have been described in many papers. Their disadvantages are that they are time-consuming and labor intensive, which leads to difficulties when there are many samples to analyze [55,56,57]. Table 4 presents the inhibition zones for the studied samples. The study of the antimicrobial activity highlighted a greater inhibition potential for samples P2 and P4. Between these two samples, P2 exhibited increased inhibitory activity for all types of MO, especially for unicellular fungi and Gram-positive bacteria. On the other hand, sample P4 exhibited more pronounced inhibition of Gram-negative bacteria and unicellular fungi than of Gram-positive bacteria, which would constitute an advantage for this type of structure to combat Gram-negative bacterial strains.

#### 2.1.9. In Vitro Cell Studies

The cell viability of L929 cells treated with extracts from the samples P1 to P4 is reported in Table 5. For the samples P1 and P3 which do not contain turmeric, the extract did not affect the viability of L929 cells at 24 and 48 h. For the samples P2 and P4, the addition of turmeric extract induced a significant decrease in cell viability for the P2 extract, both at 24 and 48 h, compared with P1 and the control. Only a slight decrease for the P4 extract compared with P3 and control conditions was observed. Similar results were reached for the extracts after 6 days of maintaining the culture medium in contact with the developed materials. The response was assessed again at 24 and 48 h and the recorded values are also displayed in Table 5. The cell viability was slightly improved compared to the 24 h results; most values exceeded the control and even for the sample P2, that previously showed a weaker behavior, the situation was improved. Indeed, samples P1 and P3, namely the materials without the addition of turmeric extract, appeared to be superior to samples P2 and P4, but this fact can be explained based on the existence of other secondary compounds in the extracts that could have a detrimental effect on cell viability, or on the integration of a too high concentration of curcumin, above the recommended threshold. However, considering the beneficial influence of curcumin when it comes to other properties, a slight decrease in cell viability is a compromise that we can afford. The results proved that the prepared samples will not release toxic molecules when found in the environment surrounding the wound and that they can be successfully used as a base for preparing wound dressings. Our results are in agreement with previous data reported for methylcellulose-based composites loaded with curcumin that were reported by Roşu et al. (2017) [53]. Simion et al. (2016) studied curcumin nanoemulsions and also concluded that higher curcumin concentrations exhibited increased toxicity in human endothelial cells. [58]. Other recent studies which also support our findings are those of Sajjad et al. (2020) and Chiaoprakobkij et al. (2020) [19,59]. In both studies, it was demonstrated that curcumin can induce an increase in cell viability at smaller concentrations as compared with higher ones [19,59].

Based on our results, we can say that all the prepared samples (P1–P4) have the potential to be used as wound dressings. Considering the formulation used was not toxic when placed in the growing media, the shape and structure of the films can be tailored to a specific design in the future and further studies regarding the stimulation of fibroblast cell growth can be investigated.

After analyzing all the experimental data presented in this study, we believe that the main objectives were achieved. The new composites containing BC–CMC with TE have a high swelling degree in PBS medium, namely more than 50 g/g for composites in which CMC was crosslinked with the lowest concentration of CA. The antioxidant activity was moderate in comparison with pure curcumin; however, considering previous studies, even if an increase of the curcumin content of the sample would increase the antioxidant capacity, the cell viability would be negatively affected. The samples which contained TE also had a good inhibition potential for the tested microorganisms. The in vitro cell studies proved that our samples with TE maintained a good cell viability and were not toxic when placed in the growing media. One of the drawbacks of our samples loaded with TE was the burst effect observed in the release studies, which must be avoided in the case of wound dressing materials. In order to fix this, we have previously tested the use of multiple layers, but also other solutions could be found [60,61]. Any experimental study also raises new questions for the studied composites; therefore, in the aftermath of our study, it is necessary to test new crosslinking agent concentrations and to find the optimal concentration of curcumin which must be released.

## 3. Materials and Methods

### 3.1. Materials

All reagents used were of analytical grade and all solutions were prepared using deionized water. The following reagents were purchased from Sigma-Aldrich Chemie GmbH (Taufkirchen, Germany): sodium hydroxide (CAS-No-1310-73-2); fructose (Cas-No-57-48-7); sodium carboxymethyl cellulose Mw = 90,000 (CAS-No-9004-32-4); curcumin (CAS-No-458-37-7); 2,2-Diphenyl-1-picrylhydrazyl (DPPH) (CAS-No-1898-66-4); phosphate-buffered saline; (S)-Trolox methyl ether (CAS-No-135806-59-6); citric acid (CAS-No-77-92-9); ethanol (p.a) (CAS-No-64-17-5); YPD Agar; nutrient agar; LB broth (Miller); and LB agar (Miller).

### 3.2. Synthesis of Bacterial Cellulose

The strain employed to produce the cellulose was *Gluconacetobacter saccharivorans* isolated from apple vinegar. The inoculum was prepared by growing *Gluconacetobacter saccharivorans* at 30 °C using a rotary shaker for 3 days. Then, the inoculum was transferred to 250 mL Erlenmeyer flasks in fermentation medium in a 1:10 ratio. The fermentation was performed in a modified Hestrin–Schramm (MHS) medium containing 2% fructose as the carbon source. After 7 days at 27 °C, the pellicles formed in static culture were removed from the fermentation medium. Then, the obtained pellicles were treated with 0.5 N NaOH aqueous solution at 90 °C to remove bacterial cells from the BC membrane. The pellicles were then washed several times with distilled water until the water pH became neutral. The BC pellicles were used as wet membranes.

### 3.3. Preparation of BC–CMC–TE Composites

The dried turmeric powder (*Curcuma longa*) used in this study was produced in Madagascar and was purchased from a local market in Bucharest. The turmeric powder was extracted in ethanol (p.a), the ratio solid liquid being 1/20. The extraction was carried out in a Soxhlet apparatus for 6 h. Then, the extract was filtered and the solution was analyzed to determine the curcumin content using an UV–VIS spectrometer (UniSpec2 Spectrophotometer), at a fixed wavelength of 429 nm, according to the standard curcumin calibration curve. The curcumin content of the extract was also determined using UHPLC.

A solution of 0.2% CMC was prepared, and a known amount of CA was added (for the samples P1 and P2 the ratio CMC:CA was 1:0.3 and for samples P3 and P4 the ratio was 1:0.5). The solution was stirred at 40 °C for homogenization. BC membranes in wet state were immersed in 200 mL of the CMC–CA solution and then TE was added (20 mL for each sample). The samples were left overnight on an orbital shaker to allow the CMC solution and the turmeric extract to diffuse into the BC membranes. The samples were partially dried at 40 °C in a food drier for 4 h and were then subjected to 80 °C for 10 min for curing. Before being used, all the samples were preserved at low temperatures in a refrigerator in a polyethylene foil. The membrane thickness only was controlled by the drying process; the drying was performed between parallel plates with holes, and the initial membranes having practically the same initial thickness. For practical applications, it is necessary very carefully control the membrane thickness, especially for tissue engineering applications. This is why we consider it necessary to modify the drying molds for future applications [62].

### 3.4. Ultra-High-Performance Liquid Chromatography Analysis (UHPLC)

The turmeric ethanolic extract was analyzed by an Agilent Technologies 1290 Infinity UHPLC system consisting of a binary pump, vacuum degasser, column oven, temperature-controlled autosampler, and diode array detector (DAD). Chromatographic separation of the curcuminoids was performed on a Zorbax SB-C18 RRHD reversed phase column (2.1 mm × 100 mm, 1.8 μm, Agilent, Santa Clara, CA, USA). An isocratic mobile phase of 0.1% formic acid: acetonitrile 0.1% formic acid (65:35 *v*/*v*) with an elution volume of 0.35 mL/min and a detection wavelength of 425 nm was used. The column temperature was maintained at 50 °C and 3 µL of the sample were injected. The solutions were filtered through a 0.2 nm RC membrane prior to injection. The Agilent ChemStation software was used for data acquisition and processing.

### 3.5. Characterization of Composites P1–P4

The final samples were characterized from a thermal, compositional, structural, morphological, and biological point of view.

The morphology was investigated by scanning electron microscopy (SEM) with a FEI Quanta Inspect F50 microscope (Thermo Fisher Scientific, Waltham, MA, USA) equipped with an energy-dispersive X-ray spectroscopy (EDX) probe; the working distance was set at 10 mm and an accelerating voltage of 20 kV was used. The samples surface was coated with a thin layer of gold by DC magnetron sputtering for 40 s.

The chemical bonds and groups were studied through attenuated total reflection Fourier-transform infrared (ATR-FTIR) spectroscopy with a Nicolet iS50 spectrophotometer (Thermo Fisher Scientific, Waltham, MA, USA), in the wavenumber range of 400–4000 cm^−1^, 4 cm^−1^ resolution, and 64 scans/sample.

The crystal structure and phase composition were studied by X-ray diffraction (XRD) with a PANalytical Empyrean diffractometer (Malvern Panalytical, Almelo, Netherlands), using Ni-filtered Cu Kα radiation (λ = 1.54 Å) in the 2θ range 10–60°, with a 2°/min scan speed, 0.02 ° step size, and 0.6 s preset time. The X-ray diffraction patterns were analyzed by the Pawley method [63] in HighScorePlus 3.0.e software. Therefore, a Caglioti profile function and appropriate background functions were refined for the experimental X-ray diffraction patterns [64]. The parameters used in the refinement were linear zero shift in 2θ angles and four coefficients for the background polynomial. The resulting fitted patterns were used for crystallinity determination using the following equation:(2)$$Crystallinity\;\left(\%\right)=100\cdot \frac{\sum I_{net}}{\sum I_{total}-\sum I_{constant\;background}}$$
where: *I_net_* is the net intensity of the crystalline peak, *I_total_* is the total intensity measured in the pattern, and *I_constant background_* is the subtracted intensity of the background.

The thermal analysis TG-DSC for the samples was performed with a Netzsch STA 449C Jupiter apparatus. The samples were placed in an open crucible made of alumina and heated at 10 °C/min from room temperature up to 900 °C, under the flow of 50 mL/min dried air; an empty alumina crucible was used as a reference.

### 3.6. Swelling Studies

The P1–P4 samples were cut into square pieces with an area of 1 cm^2^. These samples were immersed in phosphate-buffered saline solution (PBS) at pH 7 at room temperature. The swollen samples were withdrawn and weighed in the wet state after removing the excess PBS solution using a filter paper. The soaking process was carried out for 200 h and the samples were weighted at different time intervals (8 or 12 h). The swelling degree was calculated with Equation (3):(3)Swelling rate (gg)=Wt−W0W0
where: *W_t_* is the weight of the swollen sample at the time t and *W*_0_ is the initial weight of the dried sample.

### 3.7. Kinetics of Curcumin Release

The release behavior of curcumin from the P1–P4 composites was investigated using the total immersion assay. The composite films were cut into square pieces of 1 cm^2^ and were immersed in 50 mL of an ethanol–PBS (60:40, *v*/*v*) mixture. The experimental vials were shaken at 37 °C at 150 rpm. From time to time, 1 mL of the solution from each vial was removed to determine the curcumin content. Fresh ethanol–PBS mixture was added to the release medium to keep sink conditions. The curcumin content was measured using a UV–VIS spectrometer (UniSpec2 Spectrophotometer, LLG-Labware, Meckenheim, Germany), at a fixed wavelength of 429 nm, according to the standard curcumin calibration curve. The calibration curve was obtained by measuring the maximum absorbance at 429 nm for different concentrations of the curcumin solutions (2–20 μM) in the ethanol–PBS mixture (60:40, *v*/*v*). To determine the equilibrium curcumin concentration, considered as the maximum value concentration which could be released for each sample, the same experiments were done for 24 h. The cumulative release of curcumin was calculated with Equation (4):(4)$$Curcumin\;release\;\left(\%\right)=\frac{M_{t}}{M_{e}}\times 100$$
where *M_t_* is the amount of curcumin released at time t and *M_e_* is the amount of curcumin released at equilibrium. The experiments were performed in triplicate and the average values were reported.

### 3.8. Antioxidant Capacity

The antioxidant activity of the samples was determined using the DPPH antioxidant assay, as reported by Akter et al. 2019 with appropriate modifications: 2,2-diphenyl-1-picrylhydrazyl was used as the free radical, Trolox as the positive control, and ethanol as the solvent [52].

The P1–P4 samples were weighed (0.1 g) and 20 mL ethanol p.a was added. The solution was vortexed (V Series Stuart Vortex Mixers) for 10 min. A volume of 400 μL was mixed with the ethanolic solution of DPPH and was left in the dark at room temperature for 30 min. The absorbance was read at 517 nm using a UniSpec2 Spectrophotometer. The TEAC equivalent concentration (μg/mL) was determined from the calibration curve and the antioxidant efficiency (AE) of the samples was calculated in comparison with a control sample with Equation (5):(5)$$\mathrm{AE},\%=\frac{Abs_{0}-Abs_{p}\;}{Abs_{0}}\cdot 100$$
where *Abs*_0_ was the absorbance of the blank and *Abs_p_* was the absorbance in the presence of the samples. The antioxidant activity of curcumin and turmeric powder was also determined, and these samples were used as references.

### 3.9. Antimicrobial Activity

The antimicrobial activity of the P2 and P4 films was investigated using the disc diffusion assay. The microorganisms used for this study were: *Escherichia coli* (as Gram-negative bacteria), *Staphylococcus aureus* (as Gram-positive bacteria) and *Candida albicans* (as pathogenic fungus). Every microorganism was cultivated on specific media: LBA (Luria Bertani agar) for *E. coli*, NA (Nutrient Agar) for *S. aureus,* and YEPD (Yeast Extract Peptone Dextrose) for *C. albicans*. The samples were cut into disc shapes with a diameter of 6 mm and were sterilized with UV (254 nm) for 30 min on each side. The inoculum for each strain was adjusted to correspond to the 0.5 McFarland Standard concentration measured at 600 nm. A total of 1 mL of inoculum for each strain was spread in petri plates and the excess was removed. The sample disks were placed aseptically in the plates, using pristine BC and BC–CMC composites without curcumin as controls. The disks were incubated at 37 °C for 24 h and afterwards, the inhibition zone was measured (IZ, mm). The results are presented for IZ (mm) at 24 h and reported as mean ± standard error of the mean.

### 3.10. In Vitro Cell Studies

#### 3.10.1. Extract Preparation

Similar sized shapes of the P1 to P4 samples were cut and sterilized by UV light before extract preparation. The samples were hydrated in 5 mL culture medium and kept in an incubator for 24 h and 6 days to obtain the extracts. The surfaces were weighed, and the values found were used to estimate a concentration of the extracts. Thus, we obtained a concentration of 19.67 ± 3.76 mg/mL for P1, 18.32 ± 1.23 mg/mL for P2, 14.10 ± 1.16 mg/mL for P3, and 14.95 ± 0.53 mg/mL for P4. For each condition, two different surfaces were used in order to obtain the extract and each extract was tested separately at least two times.

#### 3.10.2. Cell Viability Assay

To assess cell viability, the MTT (3-[4,5-dimethylthiazol-2-yl]-2,5-diphenyltetrazolium bromide) colorimetric assay was performed 24 and 48 h after the application of the extracts on the mouse fibroblast L929 cell line (CCL-1, ATCC). The cells were grown in DMEM (Dulbecco’s Modified Eagle Medium) with 10% fetal bovine serum (FBS) and 1% penicillin–streptomycin and kept in an incubator at 5% CO_2_ and 37 °C. Approx. 24 h before the viability assay, 7000 cells per well were seeded into 96-well plates. The cells grown in the presence of medium alone were used as a negative control. At 24 and 48 h after the application of the extracts to the cells, the culture medium was removed and the MTT solution with a concentration of 0.5 mg/mL was added. The cells were further incubated at 37 °C for 4 h. This colorimetric technique is based on the transformation of tetrazolium salt (yellow color) into formazan crystals (purple color) in metabolically active cells. Thus, after the 4 h of incubation, the formazan crystals that formed in viable cells were dissolved using DMSO (dimethyl sulfoxide). The absorbance of the samples was recorded at λ = 490 nm using a plate reader (Mithras LB 940, Berthold). All absorbance data was corrected by subtracting the absorbance of the plate. Afterwards, the cell viability was calculated using Equation (6):(6)$$\%\;viable\;cells=\frac{Absorbance\;of\;treated\;cells}{Absorbance\;of\;control\;cells}\times 100$$

All data were presented as mean ± standard deviation (SD) unless stated otherwise, from at least two independent experiments and at least four biological replicas. Statistical analysis was performed using GraphPad Prism 5.01 software (San Diego, CA, USA). The statistical significance of the differences between experimental groups was calculated using one-way analysis of variance with Tukey’s Multiple Comparison Test. The values of *p* < 0.05 were considered statistically significant.

## 4. Conclusions

The need to develop new types of materials for healing skin wounds based on biomaterials and natural antimicrobial substances led to the choice of two biopolymers (BC and CMC) and turmeric extract to produce new composite materials for wound dressings. The main idea was to obtain composite materials with improved properties in comparison with each of the two biopolymers alone. Therefore, the lack of antimicrobial activity of bacterial cellulose was compensated by the components of the turmeric extract, especially curcumin. The loss of the rehydration capacity of BC after drying was compensated by the impregnation with CMC and crosslinking with citric acid. In addition, the new composites proved to have a high swelling capacity, which is very important to absorb the exudate of the wound. These composites not only had antimicrobial and antioxidant properties, but they also lacked toxicity and showed good cytocompatibility. These findings allow us to recommend the new BC composites as promising materials for wound dressing applications. In vivo studies are necessary in the future in order to prove their healing properties.

## Figures and Tables

**Figure 1 ijms-24-01719-f001:**
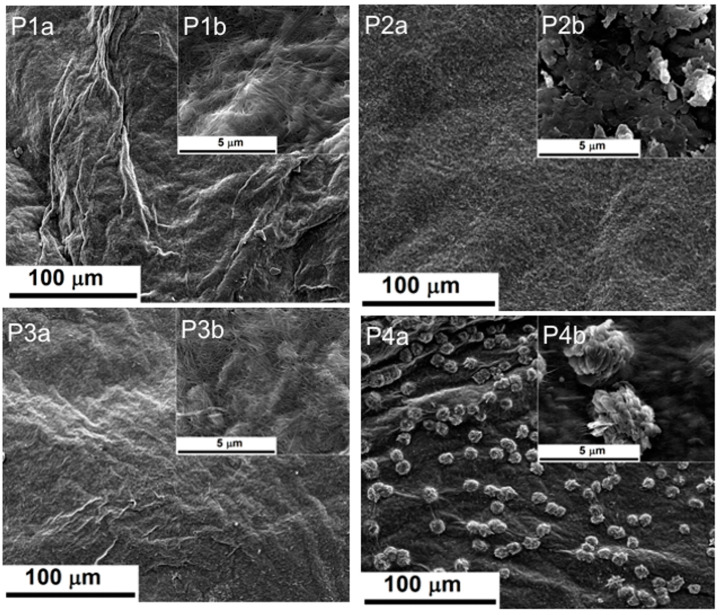
SEM images of P1–P4 samples at different magnifications: (**a**) P1a–P4a (1000×); (**b**) P1b–P4b (30,000×).

**Figure 2 ijms-24-01719-f002:**
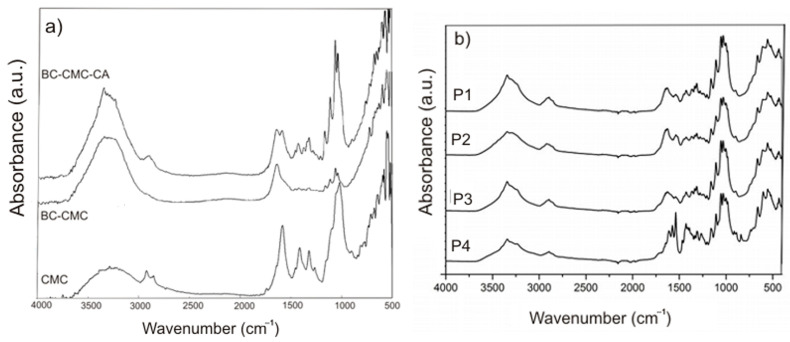
FTIR spectra of: (**a**) CMC, BC–CMC composite and BC–CMC composite crosslinked with CA and (**b**) samples P1–P4.

**Figure 3 ijms-24-01719-f003:**
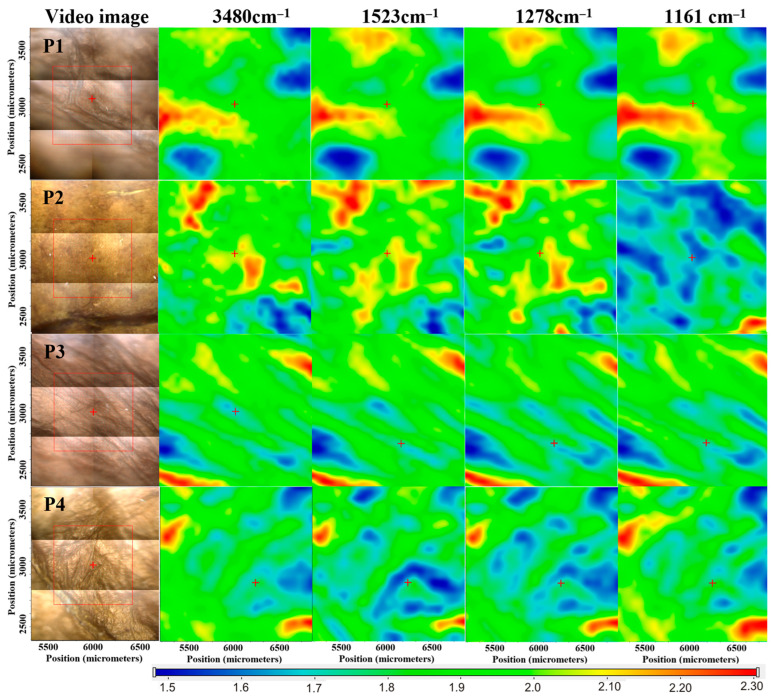
FTIR map for P1–P4 samples.

**Figure 4 ijms-24-01719-f004:**
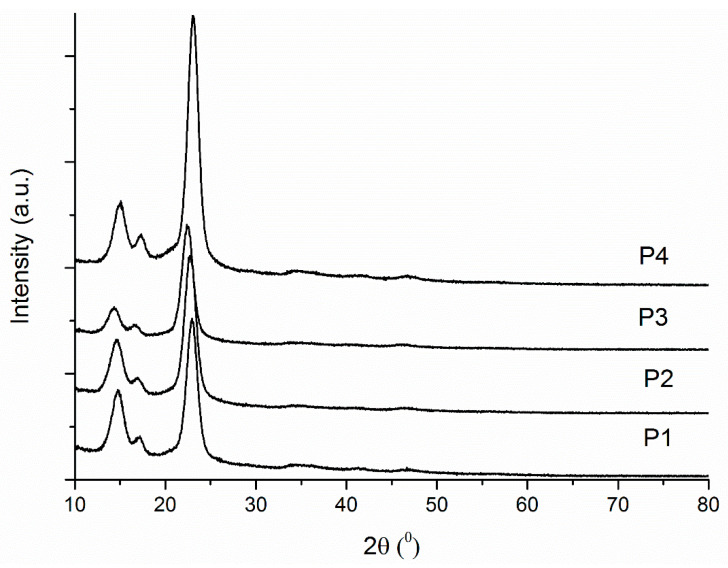
X-ray diffractograms for P1–P4 samples.

**Figure 5 ijms-24-01719-f005:**
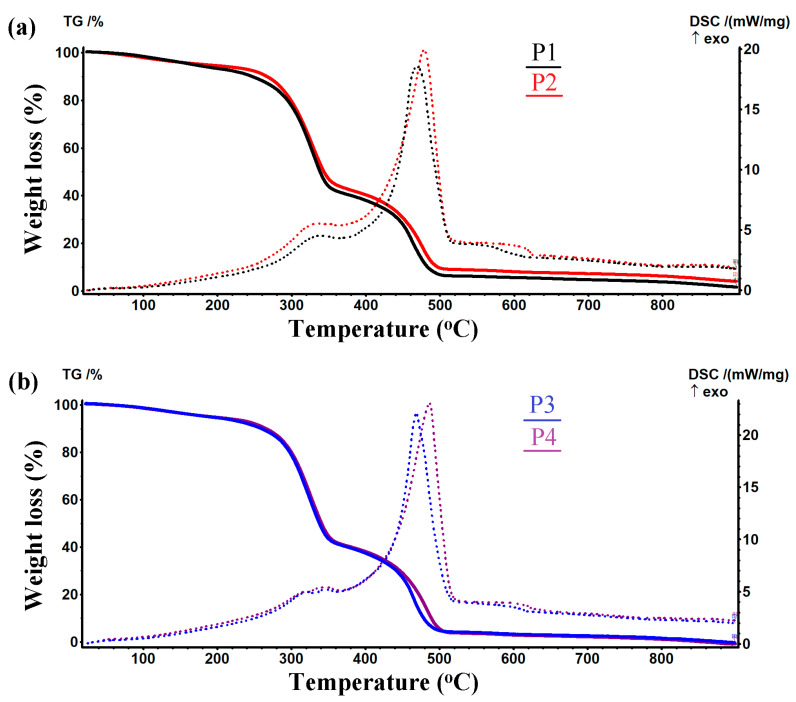
Thermogravimetric analysis (TG and DSC) of the studied samples: (**a**) P1 and P2 and (**b**) P3 and P4.

**Figure 6 ijms-24-01719-f006:**
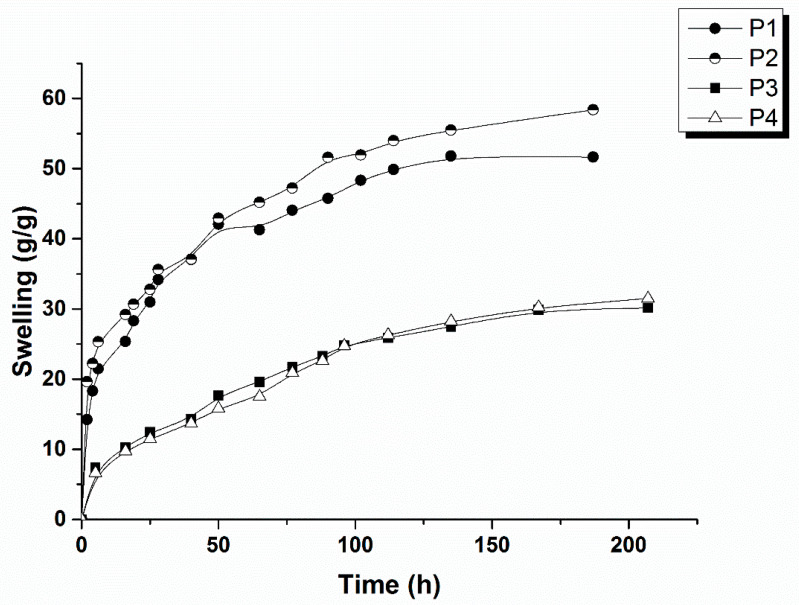
Swelling ratio of the studied samples: P1–P4.

**Figure 7 ijms-24-01719-f007:**
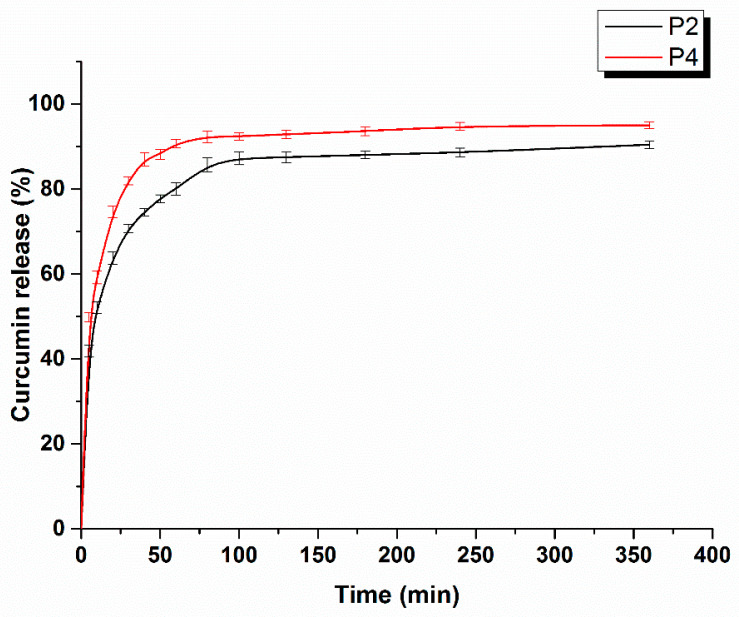
Curcumin release profile for P2 and P4 samples (*n* = 3; error bars = SD).

**Table 1 ijms-24-01719-t001:** Sample symbolic notation.

Sample Abbreviation	Description of the Sample
P1	BC-CMC-CA-1 (1:0.4:0.12 ratio) *
P2	BC-CMC-CA-1 loaded with TE
P3	BC-CMC-CA-2 (1:0.4:0.24 ratio) *
P4	BC-CMC-CA-2 loaded with TE
BC is reported as dry material.	

* BC—bacterial cellulose; CMC—carboxymethyl cellulose; CA—citric acid; TE—turmeric extract.

**Table 2 ijms-24-01719-t002:** Centralized data from thermogravimetric curves of the studied samples.

Sample	Mass Loss (%)			DSC Effects		
	RT-200 (°C)	200–375 (°C)	375–520 (°C)	* Endo I (°C)	* Exo I (°C)	* Exo II (°C)
P1	6.70	52.83	34.26	60.1	340.1	469.0
P2	5.58	51.66	33.87	60.1	334.9	478.5
P3	5.45	54.50	35.94	60.8	322.5/345.8	469.0
P4	5.32	54.23	36.79	60.0	315.5/345.7	486.6

* Endo—endothermal process; Exo—exothermal process.

**Table 3 ijms-24-01719-t003:** The antioxidant activity parameters obtained for the P2 and P4 samples, curcumin, and turmeric powder.

Sample Code	TEAC (μg/mL)	AE (%)
P2	15.49 ± 1.011	22.17 ± 0.014
P4	11.65 ± 0.93	16.81 ± 0.011
Curcumin	59.14 ± 1.025	84.65 ± 0.025
Turmeric	43.08 ± 1.02	61.65 ± 0.03

**Table 4 ijms-24-01719-t004:** Inhibition zones for analyzed samples.

Sample Code	IZ, mm ± SD Mean		
	*E. coli*	*S. aureus*	*C. albicans*
P1	0.67 ± 0.31	1.00 ± 0.41	2.04 ± 0.63
P2	1.33 ± 0.47	2.02 ± 0.33	3.67 ± 1.70
P3	0.17 ± 0.24	1.00 ± 0.27	1.94 ± 0.67
P4	1.33 ± 0.47	1.00 ± 0.29	2.44 ± 0.42
Control polymeric matrix	nd	nd	nd
Control TE extract	>20 mm	2.01 ± 0.37	nd

nd—not detected.

**Table 5 ijms-24-01719-t005:** The effect of P1 to P4 extracts on L929 cell viability after 1 day and 6 days of treatment.

Sample Code	Time		Cell Viability (%)	*p*
Control	1 day	24 h	100 ± 0.01	-
		48 h	100 ± 0.01	-
	6 day	24 h	100 ± 0.01	-
		48 h	100 ± 0.01	-
P1	1 day	24 h	99.018 ± 8.068	0.9995
		48 h	105.381 ± 2.409	0.7744
	6 day	24 h	103.631 ± 4.569	0.9926
		48 h	107.906 ± 2.973	0.7120
P2	1 day	24 h	79.426 ± 5.676	0.0046 *
		48 h	82.368 ± 3.332	0.0130 *
	6 day	24 h	100.294 ± 14.233	0.9999
		48 h	98.089 ± 12.595	0.9978
P3	1 day	24 h	102.383 ± 11.305	0.9859
		48 h	106.241 ± 8.770	0.6798
	6 day	24 h	106.975 ± 18.176	0.9224
		48 h	103.738 ± 4.004	0.9728
P4	1 day	24 h	95.755 ± 0.531	0.8947
		48 h	95.805 ± 11.062	0.8917
	6 day	24 h	102.232 ± 13.466	0.9989
		48 h	105.923 ± 14.269	0.8719

* Statistical significances between samples are expressed as *p* < 0.05.

## Data Availability

Not applicable.

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
