# Peer review of "Bacterial Cellulose—Carboxymethylcellulose Composite Loaded with Turmeric Extract for Antimicrobial Wound Dressing Applications"

_ijms, 2023, doi:10.3390/ijms24021719_

Round 1

Reviewer 1 Report

I think authors require more investigations for the publication of their works. Some controls are missing, and the antimicrobial along with the antioxidant analyses did not calculate the IC50 (requiring the analysis in concentration variation). Data from the results/discussion do not match those in the protocol. Also, the preparation of CMC-BC-AC-TE does not indicate the formation of CMC—BC crosslinking (detailed in my comment). In conclusion, authors should really work on both on the manuscript writing and additional lab experiment before their work could be considered for publication.

Majors

1.     I could not get the idea of this research combining CMC and BC since both of the polymers are celluloses (the first is cellulose derivative). If CMC could enhance the properties of BC, why don’t you just modify BC into CMC?

2.     BC has not been confirmed. Characterizations should be carried out on BC and compared with the commercial bacterial cellulose.

3.     Authors prepare CMC-CA solution first before added into BC. Don’t you think the crosslinking would occur between CMC—CMC not CMC—BC? Therefore, the composite is not crosslinked, but it is just a physical entrapment of CA-crosslinked CMC into BC.

4.     How is the membrane thickness controlled? By using the cleaned pellicles as BC membrane, I don’t think it is possible. Moreover, after the wet membrane obtained, what medium/solution authors used to store the membrane?

5.     During characterizations, did authors firstly prepare the dried powder of the membrane? I think it is more important the pre-treatment procedure performed before the analysis, rather than the characterization protocols themselves.

6.     Table 1 should present the composition for BC, CMC, CA, and TE. Also, kindly provide the definition of the abbreviation in the table footnote.

7.     Swelling degree is measured with 7-days soaking time, but the data presented show the swelling behavior for max 200 h. That means the swelling is investigated for >8 days. Could authors clarify this inconsistency? Also kindly state the time interval of the weight measurement during the swelling degree test.

8.     Table 3. Curcumin and turmeric are presented in the table. These samples are not found in the methods. Please be consistent between the results and the research protocols. Also, AE is not sufficient, please calculate the IC­50.

9.     Positive control is missing from the antimicrobial test. Also, Table 4 should be analyzed with statistical analysis (post hoc Tukey?). Also, please calculate the IC50 not only the inhibition zone.

10.  P2 had significant inhibitory effect against the cell growth. Moreover, I don’t think the Figure is a good idea to present the data since the reader will be confused how the comparison is carried out. It’s very hard to understand what P2 (24&48h) compared with. Also, I think it would be a good idea to compare not only with control, but between cell viability after 24- and 48-h incubations. I recommend the data presentation in a table.

11.  In thermal analysis. Where is the crosslinkage cleavage and functional groups cleavage? The thermal phenomena are not described well (even too little for such complex polymer). Also, in DSC thermogram where is the endothermic peak? I could not observe it clearly, please give the label in the figure.

12.  XRD analysis should have crystallinity calculation. Kindly refer to DOI: 10.1007/s13369-022-06786-6

Minors

13.  Title. Consider antimicrobial instead of antibacterial since authors investigated C. albicans as well.

14.  Abstract. The methods should be trimmed shorter. The results should be added in concise manner.

15.  Line 28-29. “Escherichia coli (E. coli), Staphylococcus aureus (S. aureus), and Candida albicans (C. albicans)” should be italic.

16.  Line 107—117. The crosslinking explanation is superficial. Add this: The carboxylate moiety of citric acid reacts with hydroxyl moiety of both BC and CMC and forming crosslinkage between the two molecules.

17.  Add 3.1 Materials which includes all the materials used in the study. Information on where they are purchased from should also be mentioned there.

18.  Line 356. “Curcuma longa” should be italic.

19.  Define abbreviation ‘CA’

20.  Line 401. “1cm2” should be superscipted

21.  Eq. 4 should be “100%”

22.  Line 303—306 the sentence is too confusing, kindly paraphrase for better clarity.

23.  Similar results were also obtained for extracts obtained after 6 days (data not shown)” Why not shown? Authors may at least provide the data to peer-reviewer.

24.  “p < 0.05” consider p< 0.05 with italic ‘p

25.  Table 4. Instead of IZ, the column heading should be expanded ‘Inhibition zone’ also consider ‘Inhibition zone, Mean±SD (mm)’.

26.  Line 89—90. “the antimicrobial activity of BC” requires citation, I suggest: Harahap et al. 2022 J Adv Pharm Technol Res 13(3):148—153.

27.  “DPPH (2,2-diphenyl-1-picryl-hydrazyl-hydrate) free radical method is an antioxidant assay based on electron-transfer that produces a violet solution in solvents such as ethanol. This free radical, stable at room temperature, is reduced in the presence of an antioxidant molecule, causing the discoloration of the ethanolic solution” Remove this theoretical statement from methods.

28.  Line 247—248. ”For the moment, an explanation could be offered regarding SEM images” What authors meant by this statement? So unclear.

29.  “equation 1” the initial letter should be capital.

30.  Table 2. Exo and Endo should be defined in the table footnote. Also the unit should be typed in the bracket “(°C)”

31.  “oxidative-degradative process of the polymeric chains” what polymeric chains? Cellulose backbone?

Author Response

Dear Sir/Madam,                  

I would like to thank you for your efforts to review our submission and for all the useful observations that you have made, which helped us to enhance our work.

Thank you for your comments!

Yours sincerely,

PhD Anicuta Stoica-Guzun on behalf of the authors

Reviewer 2 Report

Dear Editor

The manuscript entitled “Bacterial Cellulose – Carboxymethylcellulose Composite 2 Loaded with Turmeric Extract for Antibacterial Wound Dress- 3 ing Applications”, main concerns include:

1.   Molecular quantification and characterization of Turmeric extract for wound dressing is necessary.

2.   The effect of the employed crosslinker (citric acid) on the Turmeric extract should be discussed.

3.   The novelty should be highlighted as there are a lot of already published papers in this field.

4.   For biomedical applications more than thermal and crystalline structure characterizations, some other analysis should be conducted (leaching/migration, in vivo applications, etc.).

5.   pH of the system and its potential effect on the wound dressing was not addressed.

6.   Carboxymetylated cellulose does not recommended for wound dreeseing for the proved reasons. Bacterial cellulose as the purest form of cellulose seems more viable.

7.   Some references do not back the claims and justifications needed.

8.   The introduction lacks a strong background study. The potential findings of other potential research must be included in the background. Authors should use and cite other important publications that used natural extracted for antibacterial purposes and explain their shortcomings that encouraged authors to work on another approach. The last section of the introduction must clearly focus on objectives, a summary of the methodology, and a contribution to the scientific society. Additionally, highlight the novelty and scope in the introduction.

9.   Following literature could prove this manuscript;

https://doi.org/10.1016/j.carbpol.2022.119910

https://doi.org/10.1016/j.ijbiomac.2021.05.088

10.The conclusion is so primitive, rewrite it.

I would like to review the revised paper.

Author Response

(The authors gave the same response as above.)

Round 2

Reviewer 1 Report

I thank you authors for their kind responses and improved manuscript. Though authors have made necessary modifications in the manuscript, authors forget to indicate the line and page number in the response-to-reviewer file. Authors have responded very well to my minor comments, but some major comments need further addressing. Please find below my comments:

Query #1: Authors tried to reason the combination of BC and CMC. I could not get the idea why BC-derived CMC has been used for food packaging, hence cannot be used as wound dressing. Nonetheless, authors’ reasons of combining BC and CMC to get cheaper materials could be reasonable. Please include this in the introduction with more elaboration.

Query #2: I understand that authors have started the research for quite some time, where previous publication experiences have allowed authors to confirm the BC. Nonetheless, to make the paper become more comprehensive, I suggest authors to include the discussion about BC first.

Query #3: Clear.

Query #4: Explain in the text. Also recommend better research strategies to control the membrane thickness.

Query #5: Clear.

Query #6: Clear.

Query #7: Clear. But next time please do tell which line and page the modifications have been made.

Query #8: Clear.

Query #9: Clear. But do you have any recommendations to test the antimicrobial in more comprehensive manner for wound dressing materials? Please state in the recommendation.

Query #10: “..this type of graph is a very common way of presenting the cell viability values…” Just because it is usually used, does not mean it is the best way of presenting the data. I still strongly suggest author to convert the figures into tables with p-value clearly presented.

Query #11: Still not clear to me. I don’t think peak at around 60°C is an endothermic peak.

Query #12: Clear.

Author Response

Dear Sir/Madam,                  

I would like to thank you for your efforts to review our submission.

Since a new year will soon begin, we wish you <Happy New Year!> and much success in your work!

Yours sincerely,

PhD Anicuta Stoica-Guzun on behalf of the authors

Reviewer 2 Report

The manuscript improved and most of the comments are addressed, and it can be considered for publication.

Author Response

(The authors gave the same response as above.)

Round 3

Reviewer 1 Report

Thank for this revision. I'm happy to see the improved version. I have several thing need clarification, but the manuscript would not need to be sent to me. 

1. Authors have added a new paragraph explaining that CMC is a low cost material. Kindly, add the conclusion to the paragraph that CMC-BC modification could reduce the production cost of the material.

2. For table 5. No need to make a separated table column for statistical significance, p-value is enough. And significant p-value could just simply be indicated by (*) with description in the table footnote. 

3. The zoomed in DSC thermogram showing the endothermic peaks should be presented as supporting file, because the one in the manuscript is not clear.

I think everything else is clear. Congratz on your work!

Happy new year! Cheers!

Author Response

Note to the reviewers of manuscript: Ms. Ref. No. ijms-2108242
Dear Sir/Madam,                  

I would like to thank you for your efforts to review our submission and for all the useful observations that you have made, which helped us to enhance our work.

Thank you for your comments and for your assessments!

Yours sincerely,

PhD Anicuta Stoica-Guzun on behalf of the authors
